# How Viruses Use the VCP/p97 ATPase Molecular Machine

**DOI:** 10.3390/v13091881

**Published:** 2021-09-21

**Authors:** Poulami Das, Jaquelin P. Dudley

**Affiliations:** 1Department of Molecular Biosciences, The University of Texas at Austin, Austin, TX 78712, USA; pdas@utexas.edu; 2LaMontagne Center for Infectious Disease, The University of Texas at Austin, Austin, TX 78712, USA

**Keywords:** VCP, p97, virus replication, ERAD, cellular trafficking, uncoating, egress, antiviral immune response, retrotranslocation

## Abstract

Viruses are obligate intracellular parasites that are dependent on host factors for their replication. One such host protein, p97 or the valosin-containing protein (VCP), is a highly conserved AAA ATPase that facilitates replication of diverse RNA- and DNA-containing viruses. The wide range of cellular functions attributed to this ATPase is consistent with its participation in multiple steps of the virus life cycle from entry and uncoating to viral egress. Studies of VCP/p97 interactions with viruses will provide important information about host processes and cell biology, but also viral strategies that take advantage of these host functions. The critical role of p97 in viral replication might be exploited as a target for development of pan-antiviral drugs that exceed the capability of virus-specific vaccines or therapeutics.

## 1. Introduction

Interaction with host proteins is a vital step in the infectious cycle of intracellular pathogens, particularly viruses. Viruses have co-evolved with their hosts, developing ingenious strategies for hijacking and controlling host processes for efficient viral replication and propagation. A key host protein that interacts with many viruses during their infectious cycles is the ATPase p97 or valosin-containing protein (VCP) [1,2,3,4]. Also known as Cdc48 in *S. cerevisiae* [5], VCP/p97 is highly conserved across the animal kingdom from archaebacteria to eukaryotes [6] and is a member of the hexameric AAA (ATPases associated with various cellular activities) enzyme family [7]. Evolutionary conservation implies essential p97 functions throughout the cell, explaining how viruses have become addicted to the use of its ATPase activity.

Each p97 is a homohexamer (Figure 1). The monomers are comprised of an N-terminal domain with multiple protein-binding sites and two AAA ATPase domains designated D1 and D2 [4]. The ATPase domains are highly similar in sequence and structure, yet they have distinct functions [4]. The D1 domain is essential for hexameric p97 assembly, whereas the D2 domain has the primary ATPase activity [8,9]. The D1 and D2 domains are connected by a short polypeptide known as the D1–D2 linker, and a flexible N-D1 linker joins D1 to the N-terminus [10] (Figure 1A). The p97 protein serves as an essential cellular chaperone, using its ATPase activity to drive protein unfolding [3,6]. Recent data suggest that VCP/p97 is responsible for unfolding of substrates by a hand-over-hand mechanism. The substrate is engaged by the N-terminal domain and pulled into a central pore formed by the D1 domains likely through sequential interaction with monomers and exits through the central pore of the D2 domains [11,12,13] The N-terminal domain interacts with a wide array of cofactors and adapters, but a few proteins bind to the C-terminal p97 tail [3,14,15]. ATP is known to affect the position of the N-terminal domains in the homohexamer, the “up” position associated with ATP binding and the “down” position with ADP binding [3] (Figure 1B). The “up” position often determines the ability of p97 to associate with protein adapters, such as the heterodimer, UFD1/NPL4 [16]. In turn, these adapters frequently bind to viral and cellular ubiquitylated proteins, which serve essential roles in protein–protein interactions and protein turnover.

The p97 protein is a primary signaling hub for ubiquitin or ubiquitin-related modifiers, such as SUMO and Nedd8, which are present on target proteins [17,18]. Although p97 is predominantly localized in the cytosol as a soluble protein, a fraction is present on membranes of organelles, such as endoplasmic reticulum (ER), Golgi, mitochondria, and endosomes, as well as the nucleus [6]. Depending on associated co-factors [19], p97 participates in a wide array of diverse cellular functions, ranging from ER-associated degradation (ERAD), mitochondrial-associated protein degradation [20,21], the ubiquitin proteasome system (UPS) [1], DNA replication [22], DNA break repair [23,24], chromatin structure regulation [25], NF-κB activation [26,27], Golgi formation-stalled ribosome turnover [28,29], endomembrane fusion [30], and autophagy [31]. Interactions of p97 with each of these systems is mediated by different adapters [17]. The essential role of p97 in host cell functions foretells the inevitable interactions between this ATPase and viruses.

Apart from the numerous cellular functions, VCP/p97 has been shown to participate in multiple stages of virus production, including entry and uncoating, intracellular trafficking, nucleic acid replication, and egress [32,33,34]. Although VCP may not be involved in the replication cycle of every virus, this ATPase likely plays a critical part in disease pathogenesis as well as the antiviral host response [32,33,35]. In this review, we focus on how viruses effectively use the p97 molecular machine to promote infections.

## 2. Role of VCP/p97 in Early Stages of Viral Infection

Viral entry is dependent on interactions of the anti-receptor with the host cell receptor at the plasma membrane [36]. With some viruses, a conformational change occurs in the anti-receptor at neutral pH to allow direct entry of the viral nucleic acid or a subviral particle [37]. Alternatively, viral components are delivered to the cytoplasm after interaction with a co-receptor [38]. In other cases, virus particles are internalized by endocytosis, which is needed for virions to reach a low pH compartment that will trigger envelope fusion or breakdown of endosomal membranes [37]. Endosomal trafficking is known to depend on VCP/p97 [39,40], presumably to mediate endosome fusion through binding to clathrin, syntaxin 5, and early endosome antigen 1 (EEA1) [6,30]. Therefore, it is not surprising that multiple viruses are dependent on this ATPase for entry.

Members of the positive-stranded RNA-containing *Flaviviridae* family of enveloped viruses, particularly the *Flavivirus* genus [41], use p97 for early stages of viral infection [42,43] (Figure 2). This genus includes multiple viruses causing disease in humans, e.g., West Nile virus (WNV), Zika virus (ZIKV), Dengue virus (DENV), Japanese encephalitis virus (JEV), and yellow fever virus (YFV) [41,44]. Virus entry occurs through clathrin- and dynamin-dependent receptor-mediated endocytosis [45,46]. Knockdown or inhibition of the E1 enzyme UBA1 indicated that DENV requires ubiquitylation during early viral events, which likely involves endosome maturation or nucleocapsid disassembly [47]. VCP/p97 is associated with multiple adapter proteins that recognize ubiquitylated proteins for unfolding and often proteasome targeting [6]. Experiments with WNV virus-like particles (VLPs) encoding a reporter gene revealed inhibition of early steps of infection with small interfering RNAs (siRNAs) specific for VCP [42]. These same siRNAs did not affect replication of VSV [42], a negative-stranded RNA virus, which also uses receptor-mediated endocytosis for cell entry [48]. Using a replication-defective reporter virus system to study entry and pre-replication events of multiple flaviviruses, an inhibitor of the ubiquitin-activating enzyme E1 also blocked YFV, ZIKV, and WNV uncoating [43]. This study further showed that two small molecule inhibitors of p97, DBeQ and NMS-873, which have different modes of action [49,50], interfered with reporter virus expression. VCP inhibitors did not interfere if reporter RNA was transfected [43]. Such results were consistent with the idea that the p97 ATPase and ubiquitylation are required for an event prior to viral RNA replication.

In support of p97-dependent events early during viral entry, Gestuveo et al. performed a screen in mosquito cells to identify interactions with the membrane-anchored or untethered capsid proteins of ZIKV [51]. Both ZIKV capsid proteins interacted with VCP, also known as TER94 in insect cells [52]. Co-immunoprecipitation experiments confirmed the interaction. Using specific inhibitors and time-of-addition experiments, VCP was shown to be needed early during ZIKV infection in either mosquito or human cells [51]. Furthermore, silencing of the E3 ligase UBR5 [53] decreased ZIKV reporter expression, consistent with involvement of ubiquitylation. These studies provide evidence that p97 functions in the removal of ZIKV capsid protein during uncoating of viral RNA [51].

In similar experiments, a DENV reporter virus showed sensitivity to DBeQ [54], suggesting that all flaviviruses may depend on p97 for uncoating. The exact nature of the interaction of VCP/p97 with nucleocapsids is unknown, but requirements for ubiquitylation of the capsid proteins may vary [43], perhaps mediated by p97 adapters [55]. Mutation of all capsid lysines did not prevent uncoating, yet ubiquitylation can occur on non-canonical amino acids [47]. As noted above, p97 inhibition interferes with endosome fusion [30], and therefore, VCP may participate in multiple steps of viral entry and uncoating. Furthermore, the role of VCP/p97 is evolutionarily conserved since inhibitors also blocked pre-replication events of YFV in mosquito cells [43]. Recent results with JEV revealed that the VCP inhibitor CB-5083 could inhibit viral replication, both in tissue culture and in a mouse model [56]. Additional characterization indicated that the VCP inhibitor prevented JEV from leaving clathrin-coated vesicles and interfered with capsid protein degradation [56], in agreement with the idea that ATPase activity is required for uncoating of flaviviruses.

VCP also facilitates entry of at least one member of the *Togaviridae* family, Sindbis virus (SINV), in both insect and mammalian cells [52]. SINV (genus *Alphavirus*) is a mosquito-borne virus containing a single-stranded, positive-sense RNA genome with a 5′ cap and 3′ poly(A) tail [57]. Although many infections remain asymptomatic, SINV may cause Sindbis fever in humans, which is characterized by polyarthritis, rash, and fever [58]. SINV uses the conserved cellular iron transporter NRAMP as its receptor [59]. Using an RNA interference screen in *Drosophila* cells infected by a recombinant SINV expressing green fluorescent protein (GFP), VCP/p97, SEC61A (an ER membrane translocon component), the p97 cofactors UFD1L-NPL4, and 22 proteasome components were necessary for efficient SINV infection [60]. All these proteins are known to be important for ERAD [61]. Knockdown with individual siRNAs or the drug Eeyarestatin 1, which affects p97-dependent protein degradation, Sec61-dependent translocation into the ER, and vesicle transport [62], affected SINV [60]. These same treatments did not affect vesicular stomatitis virus (VSV), a negative-stranded RNA virus in the *Rhabdoviridae* family.

Similar to YFV, the p97 inhibitor DBeQ inhibited SINV infection in insect cells [60]. The entry requirement for p97 involved regulation of the levels of the SINV receptor NRAMP [59], as well as specific plasma membrane proteins, such as β1 integrin or GLUT1 [63]. In the presence of p97 inhibitors, NRAMP was redirected to lysosomes, suggesting that p97 is needed to maintain levels of the SINV receptor. Junin virus, an arenavirus that uses TfR for entry [64], was not affected by p97 inhibitors [60]. Furthermore, not all viral receptors are regulated by p97 since VCP was not required to maintain levels of polio virus receptor (PVR) [65,66,67]. The role of VCP/p97 in SINV infection was only partially bypassed by using low pH to allow fusion of the viral envelope with the plasma membrane [60]. Such data suggest that p97 participates in both entry and post-entry steps of SINV infection. This conclusion was supported by results with WNV VLPs generated by transfections to bypass early events in the presence of VCP-specific siRNAs [42].

VCP likely affects the entry of multiple viruses through endosomal trafficking. Treatment of mammalian cells with p97 inhibitors caused peri-nuclear accumulation of early endosomes but did not affect late endosomes [30]. Although the exact role of p97 in endosomal maturation remains unclear, it appears that the early endosomal marker EEA1 is monoubiquitylated [68]. The multimeric state of EEA1 is controlled by p97 but is independent of its ubiquitylation status [68], suggesting that VCP also controls the multimerization status of viral proteins, such as capsid, during viral entry.

Viral exit from endosomes during receptor-mediated endocytosis is critical for delivery of a subviral particle into the cytosol to initiate replication, including coronaviruses (CoVs). CoV infections are associated with respiratory and enteric diseases both in humans and a wide range of animals [69]. The *Coronaviridae* are a family of enveloped, positive-stranded RNA-containing viruses [69,70]. The subfamily *Coronavirinae* contain four genera (alpha-, beta-, gamma-, and deltacoronaviruses) with the highly lethal SARS CoV-2 and Middle East Respiratory Syndrome (MERS) species in the betacoronavirus genus [70]. Severe acute respiratory syndrome CoV type 2 (SARS CoV-2) is the causative agent of the CoV-induced disease described in 2019 (COVID-19) [71]. A high-throughput genome-wide RNA interference (RNAi) study using the infectious bronchitis virus (IBV) in a human lung cell line revealed that p97 is essential for coronavirus exit from endosomes [72]. Decreased levels of N protein in p97-depleted cells were rescued in the presence of MG132, a proteasomal inhibitor [73], or bafilomycin A, which blocks acidification of endosomes [74]. Additional siRNAs specific for p97 blocked entry of both the avian IBV (a gammacoronavirus) and the human coronavirus 229E (an alphacoronavirus), suggesting that VCP is required for all coronaviruses during early events [72]. After VCP depletion, the infecting virus accumulated in an early endosomal compartment. CoV spike (S) protein requires an acidic pH for the fusion between the viral envelope and endosomal membranes [72]. These data are consistent with a role for VCP in acidification needed for membrane fusion as well as viral uncoating due to proteasomal degradation of the N protein. VCP may be required for unfolding or disassembly of N protein associated with viral RNA after ribonucleoprotein release into the cytosol [72].

## 3. Viral Genome Replication and p97

Viral genome replication represents another step that is regulated by VCP/p97. Because use of siRNAs and inhibitors specific for VCP can affect multiple cell processes, several approaches have been used to show that p97 is required for replication of different alphaviruses, such as Chikungunya virus (CHIKV), o’nyong’nyong virus (ONNV), and Semliki Forest virus (SFV). Studies by Carissimo et al. have shown that the effect of p97 inhibitors on a CHIKV reporter virus occurred after entry [33]. Using a 2-plasmid replication reporter system [75,76], VCP inhibition had no effect on RNA template or viral non-structural protein (nsP) levels, but decreased RNA replication products were observed. Confocal microscopy revealed co-localization of nsPs with VCP [33]. These nsPs are required for assembly of the RNA-dependent RNA polymerase, which is responsible for viral RNA replication and transcription [77]. VCP inhibitors also affected ONNV and SFV replication in several cell types [33]. Since many positive-stranded RNA-containing viruses use double-membrane vesicles for replication [78,79] and due to the known role of VCP in endosomal morphology and trafficking [68], p97 may be required for formation of viral lipid platforms needed for RNA replication (Figure 2).

Members of the *Flaviviridae* family, including JEV, WNV, ZIKV, DENV, and hepatitis C virus (HCV), also depend on p97 for their replication. Depletion of the VCP adapter, UFD1, did not affect JEV entry but reduced viral RNA levels [56]. Transfection of JEV RNA into HeLa cells was used to bypass the entry step and revealed that viral replication was restricted in the presence of VCP-specific siRNA. Furthermore, p97 co-localized near the nucleus with nonstructural proteins NS1 and NS5 but not capsid [56]. All of the flavivirus nonstructural proteins are involved in RNA replication [80]. Immunoprecipitation of tagged forms of NS3 and NS5 with VCP only occurred in infected cells, suggesting that other viral factors were required for nonstructural protein interactions with p97. WNV genomic replication also was decreased in the absence of p97 [56]. Using p97-specific si-RNAs in combination with a DNA-replicon system to bypass viral entry, knockdown decreased WNV RNA levels [42].

Screens using DENV nonstructural protein NS4B revealed VCP as an interaction partner [81]. Because of the similarity between DENV and ZIKV NS4B, co-immunoprecipitation experiments were performed to confirm interactions between endogenous p97 and ZIKV NS4B in cells infected by two different viral strains [82]. Expression of dominant-negative VCP or chemical inhibitors interfered with ZIKV production and indicated that p97 ATPase activity was required. NS4B induced relocalization of VCP into cytoplasmic replication structures that appear to be composed of smooth ER membranes. Mitochondrial architecture also was affected by NS4B expression [82]. Although mitochondria are known to affect apoptosis [83], caspase inhibitors did not restore ZIKV replication in the presence of a VCP inhibitor [82]. Thus, VCP likely functions in the assembly of replication factories together with flaviviral nonstructural proteins but also may delay apoptosis induced by ZIKV infection [84].

Another member of the *Flaviviridae*, HCV, causes hepatocellular carcinoma in a significant proportion of the chronically infected population [85]. Purification of the HCV replicase revealed an association with VCP, and p97-specific inhibitors caused abnormal localization of the NS5A protein [86,87]. NS5A has been shown to be phosphorylated on multiple serines by cellular kinases in the context of a polyprotein precursor [88,89], and its activity is essential for production of the viral replicase complex [90]. Based on the use of a reporter vector system or in cells expressing a subgenomic replicon, which avoids effects on HCV entry, p97 inhibitors revealed that ATPase activity was required for viral RNA replication [87]. Use of pharmacological inhibitors as well as shRNA-mediated knockdown of VCP resulted in the formation of HCV nonstructural protein 5 (NS5A), but not NS3, aggregates that were resistant to denaturation by sodium dodecyl sulfate. Treatment with p97-specific inhibitors led to decreased NS5A hyperphosphorylation [87]. Al-though the precise role of NS5A remains unknown, it may serve as a scaffold, much like oxysterol-binding protein (OSBP), to mediate contacts between lipids and proteins [91]. VCP and its cofactors likely facilitate flaviviral replicase assembly and/or function in the context of ER membranes [92] (Figure 2).

Enterovirus 71 (EV71) and poliovirus (PV) are non-enveloped neurotropic viruses with a positive-sense RNA genome in the *Picornaviridae* family [93]. EV71 is the causative agent of recurring hand, foot, and mouth disease (HFMD) that usually affects infants and young children [94]. A genome-wide RNAi screen identified p97 as a cellular factor implicated in EV71 replication. Using the VCP inhibitor DBeQ, this study suggested that p97 interfered with autophagosome formation during EV71 replication, but mechanistic experiments were not performed [95]. Experiments by Wang et al. revealed that EV71 inhibits ERAD of cell substrates that use either the calreticulin/calnexin or BiP pathways. A retrotranslocation assay with a split-GFP linked to an ERAD substrate revealed that EV71 inhibited substrate dislocation from the ER [96]. Expression of the viral protease 3C, which is required for processing of polyprotein precursors, was shown to cleave the E2 enzyme Ubc6e involved in ubiquitylation of ERAD substrates. The p97 recruitment factor UBXD8 also was cleaved during EV71 infection, but the mechanism was not determined [96]. Nevertheless, the cleavage of these two proteins provides a means for ERAD inhibition [96]. Treatment of EV71-infected cells with either a p97 inhibitor or a dominant-negative expression vector inhibited virus replication, indicating that VCP ATPase activity is required. The nonstructural protein 2C is required for the formation of replication organelles (ROs) [97]. VCP/p97 co-localized with the viral protein 2C on ROs in perinuclear regions derived from ER membranes within infected cells. Therefore, EV71 disables the ERAD machinery to use VCP and other associated factors in RO assembly [96].

The participation of p97 in viral replication on cytosolic membranes is evident for multiple picornaviruses. A high throughput screen with an siRNA library targeting membrane-trafficking proteins identified p97 as a host factor required for poliovirus replication after initial viral protein synthesis [34]. VCP was shown to interact with viral proteins 2BC and 3AB, which are required for RO assembly and priming of newly synthesized viral RNA on cellular membranes, respectively [98]. Knockdown of VCP strongly inhibited PV genome replication but had no effect on the replication of Coxsackievirus B3 (CVB), which is also a member of the genus *Enterovirus* [34]. In contrast, depletion of p97 enhanced the replication of Aichi virus, a member of the *Kobuvirus* genus. Since a mutant within the 2C region of PV was less sensitive to VCP knockdown [34], it seems likely that p97 ATPase activity is involved in RO assembly for PV replication, whereas this role may be fulfilled differently for other members of the *Picornaviridae* family. One possibility is that the 2C protein itself has the requisite ATPase activity for RO assembly of some picornaviruses [99].

The RNA-containing virus, Hepatitis E virus (HEV), which is a member of the *Hepeviridae* family, interacts with p97 [100]. Like picornaviruses, these viruses are small, non-enveloped viruses that replicate in the cytosol [101]. Pulldown experiments with the viral RNA-dependent RNA polymerase or the viral RNA identified VCP [100]. Although much less is known about HEV than PV, it is likely that the *Hepeviridae* use VCP for RO assembly and RNA replication.

VCP/p97 also functions in the replication of DNA viruses, such as human cytomegalovirus (HCMV) (Figure 3). HCMV is a member of the *Herpesviridae* family, which are enveloped viruses containing double-stranded DNA [102]. HCMV infects more than 90% of local populations depending on socio-economic status [103]. Infection typically is asymptomatic in healthy individuals but causes significant morbidity and death in immunocompromised individuals and newborns [103]. Using a focused siRNA screen directed against 160 membrane-associated factors, Lin et al. showed that knockdown of VCP had the greatest negative effect on a recombinant HCMV expressing GFP [32]. Knockdown of p97 revealed that no viral replication occurred. Subsequent experiments showed that the immediate early gene product IE2 was reduced after inhibition of VCP, but the IE1 gene product was not [32]. IE1 and IE2 are produced from the same transcript by differential splicing [104], and p97 is essential for efficient switching of splicing from exon 3 to exons 4 or 5, for IE1 and IE2, respectively [32] (Figure 3). In HCMV-infected cells, VCP also strongly relocalized to viral replication compartments in the nucleus, and use of p97 inhibitors blocked virus replication. VCP was shown to have multiple roles during HCMV infection since p97 inhibition at 24 h post-infection also inhibited viral production [32]. Although VCP has been shown to have a role in RNA processing [105], its mechanism of action during HCMV infection remains to be determined.

Consistent with multiple roles for VCP during HCMV infections, a recent study has implicated p97 in trafficking of capsids from the nucleus to the cytosol [106] (Figure 3). HCMV pUL50 protein is synthesized within the ER and traffics to the inner nuclear membrane for interaction with a second viral protein, pUL53. These proteins interact with cellular proteins to form the nuclear egress complex (NEC), which is conserved among herpesviruses [107]. Transfection of expression vectors for pUL50 protein was shown to decrease p97 levels post-transcriptionally and interfere with IE1/IE2 expression levels as well as the transition between the two proteins [106]. The transmembrane domain of pUL50 was required for p97 decline and was antagonized in the presence of the proteasome inhibitor MG132 or the lysosomal inhibitor bafilomycin A1 [106]. HCMV also encodes a 26-kDa N-terminal truncated isoform, UL50-p26, which is synthesized from an internal methionine codon within pUL50. This truncated protein competes with UL50 for binding to p97, thereby promoting the efficient expression of p97 and eventually efficient viral replication [106]. Viral manipulation of VCP by UL50-p26 appears to optimize the timing of immediate early gene expression. The mechanism of VCP downregulation by UL50 remains obscure. It would be interesting to determine whether p97 itself is ubiquitylated for proteasomal degradation using UL50 as an adapter to a transmembrane E3 ligase.

In addition to mammalian viruses, VCP has been reported to facilitate replication of the nucleopolyhedrovirus, AcMNPV, which belongs to the *Baculoviridae* family. Baculo-virus virions infect insects of the orders Lepidoptera, Hymenoptera, and Diptera and contain large, circular double-stranded DNA genomes [108]. After infection of Sf9 insect cells from the fall armyworm *Spodoptera frugiperda*, Lyupina et al. applied the reversible p97 inhibitor, NMS-873, at one-hour post-infection to bypass entry and nuclear transport of viral particles. The inhibitor treatment led to a dose-dependent inhibition of viral DNA replication, decreased viral protein levels, and virion particles [109]. Similar to herpesviruses, baculoviruses transcribe viral RNA and replicate their DNA within the nucleus [110]. The investigators proposed that the role of p97 in the baculovirus infection cycle involved the ubiquitin-proteasome system (UPS) and processing of ubiquitylated proteins within infected cells to relieve the stress associated with infection [109]. However, by analogy to herpesviruses, VCP may participate in the orderly progression of the baculovirus life cycle through disassembly of distinct viral-host complexes after entry as well as assembly (see Figure 3).

## 4. The Role of VCP in Late Stages of Viral Infection

VCP also participates in late stages of the virus life cycle. For example, p97 has been shown to be required for the release step of Rift Valley Fever virus (RVFV). RVFV is a member of the *Bunyaviridae* family, which have negative-sense, single-stranded, segmented RNA genomes [111]. This virus causes Rift Valley fever (RVF) in humans and livestock [112]. Although RVFV is mostly associated with limited febrile illness in humans, some individuals may experience hemorrhagic fever, neurological disorders, liver failure, blindness, or, rarely, death [113]. The molecular details of RVFV replication and late events that lead to disease are poorly understood.

The assembly of bunyaviruses occurs within the Golgi complex. The process is facilitated by two major structural proteins, Gn and Gc, which form oligomers that are retained within the Golgi network via the targeting signal within the Gn C-terminal sequence [114]. Vesicles containing viral proteins are transported from the ER to the Golgi complex, where the Gn and Gc oligomers are organized into the virus envelope [115,116]. Virions then bud into the Golgi lumen in virus-filled vesicles, which fuse with the host cell plasma membrane to release mature virions [116]. Screening of a library of FDA-approved compounds for activity against RVFV identified the drug sorafenib [117]. Sorafenib previously was known to cause fragmentation of the ER and Golgi complexes [118]. Moreover, this drug has been shown to prevent tyrosine phosphorylation of VCP [119]. Subsequently, Brahms et al. showed that sorafenib blocks Gn trafficking to the cell surface even when added at 8 h post-infection. Intracellular Gn levels or Gn trafficking were unchanged if the drug was added after the start of viral egress [117]. Microscopy revealed that RVFV-encoded Gn was relocalized from Golgi to the ER in the presence of sorafenib, thereby preventing viral egress [117]. Knockdown of VCP, but not a closely related AAA ATPase, NSF, reduced extracellular RVFV production and caused Gn retention within the ER, thus phenocopying the effect of sorafenib [117]. These results strongly indicate that p97 is needed for correct trafficking of RVFV glycoproteins to the Golgi complex for efficient budding.

Egress of other enveloped viruses likely are dependent on VCP. Infections of mosquitoes with several dengue strains have been shown to require the ubiquitin-proteasome system [120]. By using antibody-dependent opsonization of DENV particles to infect human monocytic cells, which are believed to be clinically relevant to disease induction, entry steps were bypassed. The results showed that viral RNA levels were not affected. Instead, egress of virions was decreased by proteasomal inhibitors [121]. Although the role of VCP was not tested in dengue egress, the critical participation of p97 in ERAD [2,61] is suggestive of its involvement (Figure 2).

## 5. Involvement of p97 in Viral Manipulation of ERAD and Immunity

ERAD is a critical cellular system for protein quality control and sterol regulation [122]. This process is conserved across all eukaryotes and involves a series of steps that result in clearance of misfolded or misassembled proteins synthesized on ER membranes [123,124]. These steps typically include: (i) detection of misfolded/unfolded proteins in the ER membrane or lumen, (ii) extraction/retrotranslocation of the targeted proteins through membrane channels, (iii) recognition and polyubiquitylation, and (iv) delivery to cytosolic proteasomes for degradation [125]. The ATPase activity of VCP provides the energy required for extraction of the misfolded proteins, thereby driving the energetically unfavorable process. Although ERAD is required for maintaining cellular homeostasis, this machinery often is hijacked by exogenous pathogens, such as viruses and toxin-producing bacteria, for successful infection and evasion of the host immune response. As discussed below, multiple members of the *Retroviridae*, *Hepeviridae*, *Polyomaviridae*, and *Herpesviridae* families exploit ERAD for their propagation (for review, see Byun et al., 2014) [61].

Human cytomegalovirus (HCMV) and murine gammaherpesvirus 68 (MHV68), members of the *Herpesviridae*, harness ERAD and VCP to degrade major histocompatibility class I (MHC-I) molecules [126,127,128]. The HCMV genome encodes two proteins, US2 and US11, which independently downregulate MHC-I molecules on the surface of infected cells. US2 associates with MHC-I and utilizes the ERAD components BiP [129], Sec61 [130], p97 [131], and the ubiquitin ligase TRC8 to promote MHC-I degradation [132]. In US2-mediated downregulation of MHC-I, p97 acts without the typical adapters NPL4 and UFD1 involved in recognition of ubiquitylated cellular targets [131], a relative rarity. In contrast, US11 targets MHC-I to the ERAD pathway for degradation using conventional ERAD players, including the chaperone BiP [129], SEL1L [133], Derlin-1 [134,135], p97, UFD1, NPL4 [136], and the tri- membrane spanning ubiquitin ligase TMEM129 [137,138]. Usage of TMEM129 by US11 differs from the typical ERAD ubiquitin ligase, HRD1 [138]. After recognition by BiP or other ER luminal proteins, Derlin-1 often is engaged for dislocation of proteins from the ER membrane. Derlin-1 is a multi-pass ER transmembrane protein that recently has been shown to be a homotetramer with a channel large enough to transfer a protein α-helix to the cytosol, i.e., to serve as a retrotranslocon [139]. Cytosolic E1 and E2 enzymes cooperate with ER transmembrane E3 ligases to ubiquitylate target proteins prior to recognition by adapters and VCP for delivery to the proteasome [124]. It is possible that US2 and US11 target MHC-I for recognition by BiP and subsequent degradation through p97 by changing MHC-I conformation or assembly.

Retroviruses also target MHC-I for ERAD using VCP [61]. Vpu, which is encoded by the lentivirus HIV-1, is known to downregulate MHC-I [140], as well as other surface molecules, such as CD4 and tetherin/BST-2 [141,142,143]. Experiments revealed that p97 is necessary for retrotranslocation and proteasomal degradation of the target proteins [144]. Both MHC-I and CD4 are involved in adaptive immune response to viruses [145], whereas tetherin serves to prevent release of newly formed virions [142,143]. Vpu is phosphorylated by casein kinase II [146] after insertion into the ER membrane and interacts with the E3 ligase complex SCF^βTrCP^ [147] to degrade MHC-I and CD4 using an ERAD-like process that is not dependent on typical transmembrane E3s, e.g., HRD1 and gp78 [61]. Using a hybrid dihydrofolate reductase (DHFR)-miniCD4 (mCD4) fusion protein, proteasomal degradation of DHFR-mCD4 was blocked by either MG132 or the p97 inhibitor CB-5083 [148]. Addition of the ligand, trimetrexate, interfered with retrotranslocation due to the rigid conformation induced in DHFR, indicating that unfolding of substrates is needed [148]. Thus, retroviruses exploit VCP and ERAD to avert antiviral host responses.

Trafficking of viral proteins represents another example of ERAD exploitation by viruses. For example, the betaretrovirus, mouse mammary tumor virus (MMTV), subverts ERAD and p97 for the trafficking of the Rem protein (regulator of export/expression of MMTV RNA) [149,150]. MMTV is a complex betaretrovirus that induces breast cancer and lymphomas in mice [149,151]. MMTV-encoded Rem is produced from a doubly spliced version of envelope mRNA and is synthesized as a 301-amino-acid precursor protein on the ER membrane in the same open reading frame as Env [149]. The Rem precursor is cleaved by signal peptidase into a 98-amino-acid N-terminal SP and a C-terminal glycosylated product, Rem-CT. A portion of uncleaved Rem is subjected to ERAD and accumulates in the presence of the proteasomal inhibitors MG132 and lactacystin [150] (Figure 4A). SP is cleaved from both Rem and Env proteins and is required for viral RNA export from the nucleus to the cytosol [149,152] as well as post-export processes likely to be involved in translation [153]. SP has nuclear import and export sequences as well as RNA-binding motifs, which allow binding to the Rem-responsive element near the 3′end of all viral mRNAs [149,153]. Trafficking of SP is assisted by VCP, suggesting that p97 provides the energy for ER membrane extraction [150,154].

MMTV-encoded SP uses VCP for dislocation from the ER membrane, yet it avoids proteasomal degradation typical of ERAD substrates prior to nuclear entry [150,155] (Figure 4A). Retrotranslocation of SP does not require conventional ERAD components, such as Derlins [154]. Moreover, despite being rich in lysine residues, SP is not ubiquitylated [154]. A high-throughput assay based on the ubiquitin-activated interaction trap (UBAIT) method coupled with mass spectrometry analysis identified VCP/p97 as the top candidate for SP binding. Surprisingly, no p97 cofactors were identified in this screen [154]. Expression of a dominant-negative p97 protein lacking ATPase function resulted in a significant reduction of SP activity in a reporter assay [154,155]. We recently developed a retrotranslocation assay based on expression of biotin ligases and SP fusion to a biotin affinity peptide (BAP) tag. Using this assay, we have shown that SP extraction requires VCP and its ATPase activity as judged by inhibition of SP retrotranslocation in the presence of the VCP inhibitor CB-5083. SP mutants that failed to immunoprecipitate with p97 lacked functional activity, as expected if SP cannot be extracted from the ER membrane (Das et al., in preparation). Based on our observations, we propose that SP interacts with the ATPase p97 directly to allow ER membrane extraction without ubiquitylation and proteasomal degradation.

Our recently published data also indicated that VCP is necessary for the trafficking of the C-terminal portion of Rem, Rem-CT [156] (Figure 4B). Rem-CT is localized primarily in the ER lumen despite lacking a retrograde trafficking signal, such as KKXX or KDEL [157]. A smaller fraction of Rem-CT is found within the ER Golgi intermediate compartment (ERGIC) [158] but does not transit the Golgi, as evidenced by its lack of complex glycans [156]. Instead, Rem-CT appears to traffic from the ERGIC to the late endosomes and then uses VCP for trafficking to early endosomes. VCP/p97 previously has been shown to participate in endosomal formation and size [68]. This unique pathway mediated by VCP may allow Rem-CT to sequester or degrade cellular proteins involved in innate or adaptive immunity (Figure 4B) [156].

VCP interacts with one or more viral proteins to alter the proteasome-mediated degradation of cellular proteins that affect the immune response. One such example is provided by the transactivation of the transcription factor, nuclear factor kappa B (NF-κB), by Hepatitis B virus X protein (HBx) [159]. Hepatitis B virus (HBV), a partially double-stranded DNA virus within the *Hepadnaviridae* family, is associated with hepatocellular carcinoma [160]. HBx protein acts as a multifunctional transcriptional transactivator of several cellular and viral genes and is associated with viral pathogenesis, including liver cirrhosis and carcinogenesis [161]. Using a CytoTrap yeast two-hybrid assay, Jiao et al. identified p97 as a HBx-interacting protein in addition to 28 other cellular proteins [159]. The HBx/p97 interaction was confirmed by immunoprecipitation and pull-down assays [159]. VCP/p97 is a stimulator of the transcription factor NF-κB [27], which under normal circumstances is sequestered in the cytosol in a complex with the inhibitory κB protein (IκBα) [162,163]. The IκB proteins mask the nuclear localization sequences of NF-κB proteins for their retention in the cytosol. After appropriate cell signaling, IκBα is phosphorylated and ubiquitylated, allowing association with p97 to mediate proteasomal targeting [164]. Degradation of IκBα frees an NF-κB heterodimer (often p50/RelA) for nuclear translocation and stimulation of transcription [165]. Although the enhancement of NF-κB activity by VCP overexpression in the presence of HBx was relatively small [159], the basal levels of VCP are high, and this study requires confirmation by use of a p97 knockdown or expression of a dominant-negative protein. Nevertheless, the known function of VCP in the assembly and disassembly of multiple protein complexes [136] suggests that this ATPase participates in control of viral transcription as well as activity of transcription factors, such as NF-κB, which function in both innate and adaptive antiviral immunity.

Virus-specific antibodies can bind to the capsids of non-enveloped viruses, such as the double-stranded DNA virus, adenovirus [166]. Entry of these opsonized virions into the cytosol leads to recognition by the cytosolic E3 ligase, TRIM21 [167], a process known as antibody-dependent intracellular neutralization (ADIN) [35]. Depletion of TRIM21 blocked ADIN [168]. ADIN relies mostly on the ability of TRIM21 to bind to the Fc domain of IgG, IgA, and IgM with high affinity [169,170]. TRIM21 is recruited to antibody-bound virions in the cytosol. Hauler et al. have shown that VCP is required for TRIM21-mediated intracellular neutralization by depletion of p97 with siRNA [171]. Inhibition of proteasome activity using DBeQ also prevented ADIN, but the inhibitor had no effect on adenovirus replication in the absence of antibody [171]. As described earlier, the ATPase function of VCP drives disassembly and/or unfolding of viral capsids in uncoating [43]. However, with the participation of TRIM21 and antibody, capsids may be prematurely uncoated and delivered to the 20S proteasome core via the 19S regulatory particle [172]. VCP appears to recognize and unfold highly structured complexes, whereas the proteasome requires an unfolded portion of its substrate [173]. During ADIN, both antibody and capsid proteins are degraded within a few hours post-infection [171]. Premature uncoating of viral nucleic acids can allow access to nucleic acid sensors for RNA or DNA in the cytosol [174]. Therefore, VCP may facilitate viral uncoating or promote innate immune sensing and limited viral replication in the presence of an adaptive immune response.

## 6. Conclusions

Besides its multifaceted role in regulating cellular homeostasis, the VCP/p97 ATPase serves as an important host factor in viral infections. VCP, primarily via its enzyme activity, functions in multiple stages of the viral infectious cycle, including receptor binding and entry, replication, and viral egress. Furthermore, viral manipulation of both innate and adaptive immune responses using VCP contributes to chronic infections and virus-induced diseases. Despite multiple studies, the molecular details underlying p97 participation in each of these processes is not completely understood. Nevertheless, the development of new and less toxic p97 inhibitors for clinical applications [175] provides enormous opportunities for antiviral treatments, particularly with respect to pathogenic viruses that use VCP at several different steps of their replication cycles. The requirement for p97 in numerous host processes may limit both the dose and length of treatment with VCP inhibitors. On the other hand, targeting of a host factor provides little opportunity for viral escape mutations and a potential means for a pan-viral therapeutic.

## Figures and Tables

**Figure 1 viruses-13-01881-f001:**
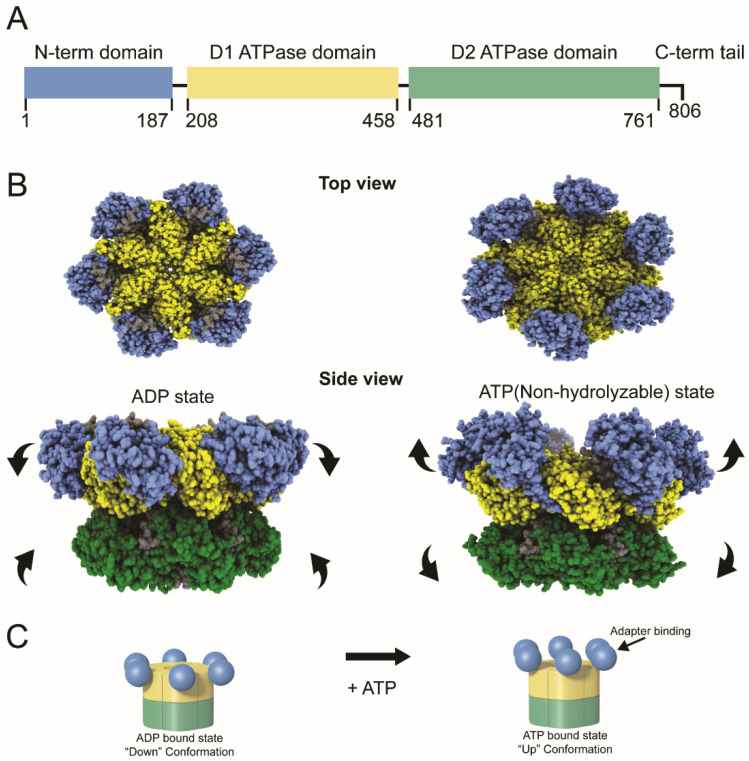
Domain structure and conformational changes in the VCP/p97 ATPase. (**A**) Diagram of the domains within the VCP/p97 monomer. Amino acid positions of each domain are indicated. (**B**) Cryoelectron microscopic images of VCP hexamers in the presence of ADP and ATP (non-hydrolyzable analog). The top view shows the propeller-like appearance of the six monomer subunits surrounding a central channel. The side view depicts the conformational changes of each monomer in the presence of ADP or ATPγS. The N-terminus and D1 and D2 domains are depicted in blue, yellow, and green respectively; other regions are shown in gray (PDB:5ftl and PDB:5ftn). [3]. In living cells, the unfolding ability of VCP is likely to be dependent on the sequential conformational changes in adjacent monomers to pull the substrate through the central pore (**C**) Cartoon depiction of the conformational switch in the N-terminal domains within the p97 hexamer. Color coding of the domains corresponds to panel A. In the ADP-bound state, the N-terminal domains primarily are in the “down” conformation. In the ATP-bound state, the N-terminal domains are in the “up” conformation for interactions with other proteins, often with adapters that recognize polyubiquitylated proteins. It is likely that many VCP hexamers have a mixture of “up” and “down” conformations within monomers, which allows the hand-over-hand unfolding and extraction mechanism.

**Figure 2 viruses-13-01881-f002:**
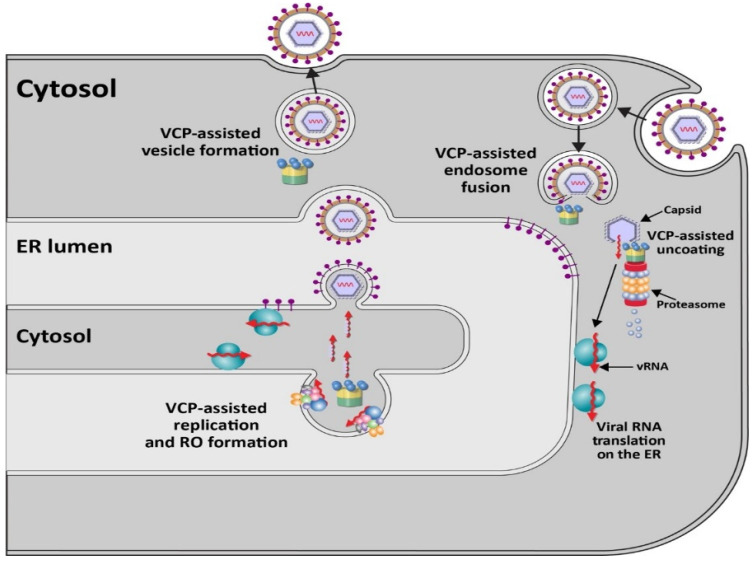
Role of VCP/p97 in the replication cycle of RNA-containing viruses. Since many RNA-containing viruses do not need the nucleus for their infectious cycles, this figure shows only a small portion of the cytoplasm, including an enlarged segment of the endoplasmic reticulum with a replication organelle (RO). Ribosomes are shown in blue, whereas viral RNAs are shown as red arrows. Viral nonstructural proteins are shown spanning the ER membrane in the RO. VCP is depicted as in Figure 1. Roles for VCP in flaviviral replication events have been documented for endosome fusion, RNA genome uncoating, and RO formation, although p97 participation in virion egress is not well documented. Other RNA-containing viruses have additional roles for p97 during viral replication. See text for details.

**Figure 3 viruses-13-01881-f003:**
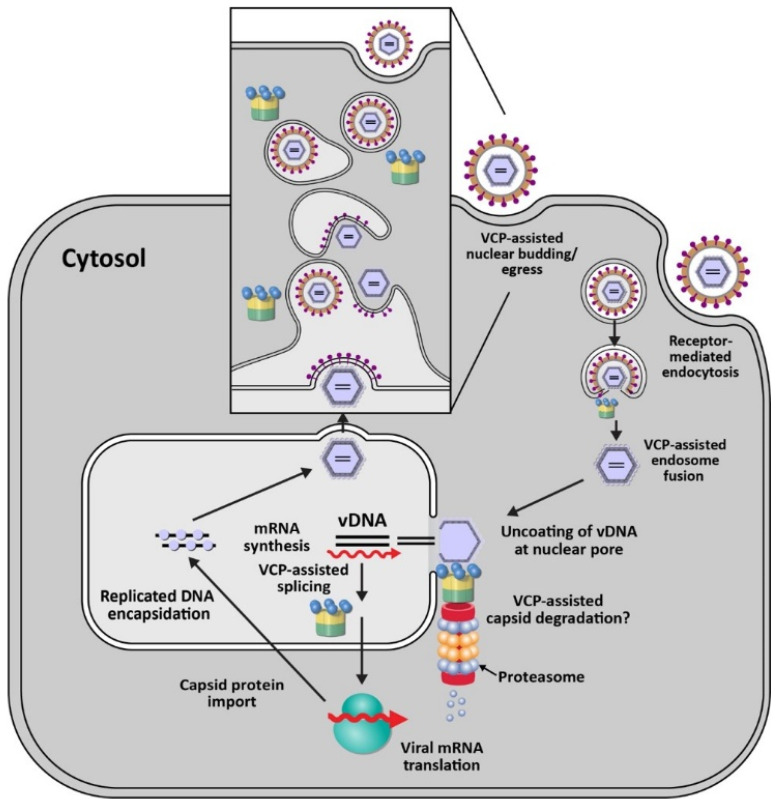
Role of VCP/p97 in the replication cycle of DNA-containing viruses. This figure is based on information derived from herpesviruses, particularly HCMV. VCP is depicted in the “up” conformation as shown in Figure 1. The DNA genome is depicted as two horizontal black lines, whereas viral mRNAs are shown as red arrows. The purple spheres on viral DNA represent capsid proteins. The inset reveals an enlarged view of viral egress from budding at the nuclear membrane, vesicle trafficking, and fusion of vesicles containing virus at the cell surface. Involvement of VCP in viral mRNA splicing and virion egress has been confirmed, but a role for p97 in uncoating is speculative.

**Figure 4 viruses-13-01881-f004:**
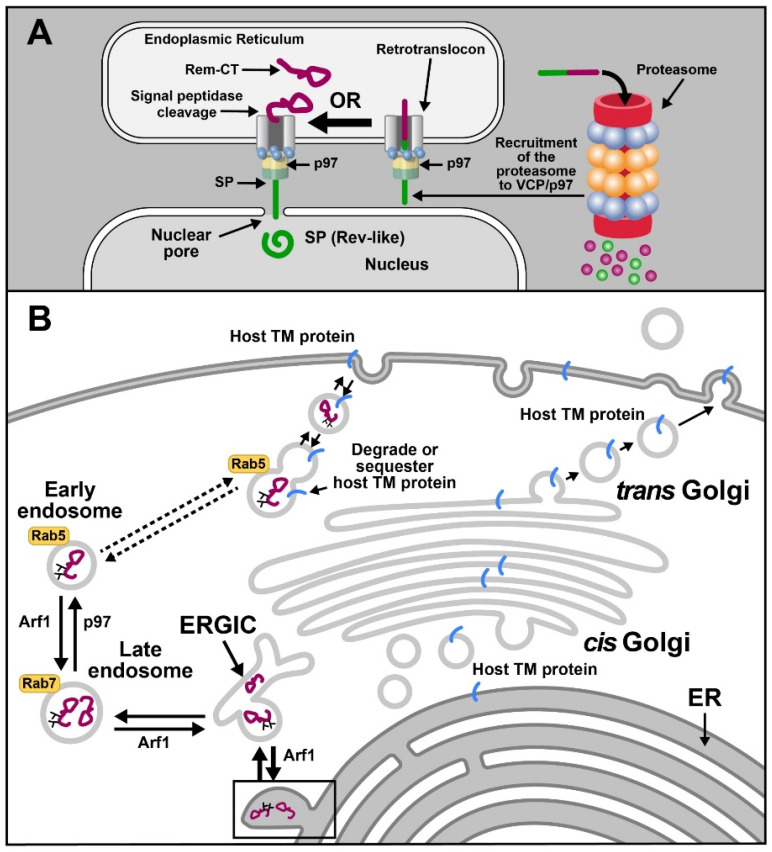
VCP/p97 and its function in retroviral protein trafficking. (**A**) Processing of the MMTV-encoded Rem protein. Rem is synthesized as a precursor at the ER membrane. A portion of Rem is extracted through a retrotranslocon by VCP/p97 and is degraded by the proteasome. The involvement of VCP adapters in this process is unknown. The majority of Rem appears to be cleaved by signal peptidase into a functional HIV-1 Rev-like protein, SP (shown in green), and a C-terminal portion, Rem-CT (shown in magenta). SP requires p97 for extraction from the ER membrane and then traffics to the nucleus to bind MMTV RNA. SP binding is necessary for efficient viral RNA export and expression. (**B**) Trafficking of Rem-CT. Rem-CT is glycosylated in the ER (Y-shaped structures) and then emerges through ER exit sites (ERES) prior to localization to the ERGIC using the activity of Arf1. Arf1 also is required for trafficking to the Rab7-positive late endosome compartment. Further anterograde trafficking of Rem-CT to the Rab5-positive early endosomes requires VCP/p97. Rem-CT localization to early endosomes or trafficking to the cell surface may allow sequestration or degradation of host transmembrane proteins that participate in innate or adaptive immunity directed against viral infection.

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
