# Peer review of "How Viruses Use the VCP/p97 ATPase Molecular Machine"

_viruses, 2021, doi:10.3390/v13091881_

Round 1

Reviewer 1 Report

This review by Das and Dudley is a well written and comprehensive work on how viruses use VCP/p97. The review is well structured and cites a wide range of the appropriate literature. Figures could be improved.

I have the following minor comments:

The role of p97 could be better described in protein unfolding (line 37).

Figure 1 could be improved with an additional panel showing X-ray or cryo EM structure of p97, moreover ’up’ and ’down’ conformations could also be described in more detail.

Figure 2 could be enlarged and ribosomes should be coloured with a solid colour.

In line 186-187 those viruses are mentioned, that do not use p97. It could already be mentioned in the introduction for better understanding.

In line 186 the abbreviation PVR should be explained.

In line 389 ’animal viruses’ should be changed to vertebrate or mammal viruses.

Literature data that is described in lines 458-478 could be better explained with a figure.

Reviewer 2 Report

This is a substantial and well written review on the role of VCP in the context of virus infections. There has been multiple recent publications on the role of VCP in viral infections indicating this review is timely and informative. A recent review on VCP touched on virus infections but not in any great detail (Med Chem. 2020 Mar 12;63(5):1892-1907). I have only minor comments.

  1. On line 335 it reads "splicing from exons 3-4 to exons 4-5" Should read axons 3-4 to axons 3-5.
  2. The conclusion is rather brief. The review may benefit from a section in the conclusion on the current VCP inhibitors and their potential as antiviral therapeutics. This was covered to some extent through out the review but perhaps commenting on the clinical potential?
